# Evaluation and Selection of Excellent Provenances of *Eleutherococcus senticosus*

Shikai Zhang [1,2], Heng Zhang [2], Luwei Ding [2], Yuxin Xia [1], Wenxue Dai [1], Xuefeng Han [3], Tuya Siqin [4,*] and Xiangling You [1,*]

[1] Key Laboratory of Saline-Alkali Vegetation Ecology Restoration, Ministry of Education, Northeast Forestry University, Harbin 150040, China
[2] State Key Laboratory of Tree Genetics and Breeding, School of Forestry, Northeast Forestry University, Harbin 150040, China
[3] Forestry and Grassland Administration of Tangyuan County, Jiamusi 154799, China
[4] Heilongjiang Institute of Atomic Energy, Harbin 150081, China
[*] Correspondence: 15904614721@139.com (T.S.); youxiangling@nefu.edu.cn (X.Y.)

**Abstract:** *Eleutherococcus senticosus* is a medicinal plant with prominent potential for pharmaceutical preparation that is naturally distributed in Northeast China. Its dried roots, stems, and fruits can be used as herbal medicine with anti-aging, anti-fatigue, anti-inflammatory, and other health benefits. With the continuous increasing of *E. senticosus* planting areas, some problems, such as a reduction in growth rate, fruit yield, and medicinal substances content, have become serious restricting factors for the long-term stable development of the *E. senticosus* industry. Therefore, the screening and selection of superior provenances, families or clones with high growth rates and high contents of medicinal substances, is urgent. In this study, 13 provenances of *E. senticosus* were analyzed and evaluated for growth, leaf, photosynthetic, and medicinal traits. The results showed that the majority of traits exhibited highly significant differences ($p < 0.01$) among provenances. The phenotypic coefficient of the variation of each trait ranged from 4.8% for leaf water content to 70.9% for total flavonoid content. The narrow-sense heritability of each trait ranged from $0.20 \pm 0.23$ for WUE to $0.39 \pm 0.14$ for DB. The variance component of all traits reached a high level, with a range of 62.9% (DB) to 99.6% (SC). The correlation analysis showed significant positive correlations between tree height, total flavonoid content, and syringin content. A multi-trait comprehensive evaluation approach enabled the selection of two provenances with 10% acceptance rate (XQ and QY). The subsequent genetic gains for tree height, total flavonoid content, and syringin content were 1.7%, 9.2%, and 20.1%, respectively. In conclusion, the present study provides a fundamental basis for the cultivation and promotion of the superior materials of *E. senticosus*.

**Keywords:** *Eleutherococcus senticosus*; genetic variation; provenance selection; growth traits; medicinal traits





## 1. Introduction

*Eleutherococcus senticosus*, also known as Siberian Ginseng, has been recognized as a traditional medicinal and edible plant species in China for centuries [1]. It is a member of the *Araliaceae* family and is native to the forests of Northeast Asia, including China, Russia, Korea, and Japan [2]. The fruits of the species are rich in various bioactive substances, such as saponins, flavonoids, coumarins, and lignans [3,4]. These substances have important health benefits, including anti-tumor [5], anti-inflammatory [6], neuroprotective [7], cardio-protective [8], antioxidative [9], anti-fatigue [10], antiedema [11] effects, and more. Among these medicinal substances, the most abundant medicinal substances are eleutherosides and flavonoids [12,13]. Eleutheroside B, also named syringin, is a phenylpropanoid compound. Experimental studies have demonstrated that eleutheroside B possesses excellent pharmacological properties, such as anti-inflammatory effects [6], analgesic and antipyretic

effects, sedative and hypnotic effects, and anti-tumor and anti-allergic effects [14]. Apart from its medicinal value, *E. senticosus* also has significant dietary benefits. For instance, it can be used to make fruit tea, natural food coloring, and milk tea [15–17]. In addition, the fresh tender bud of *E. senticosus* can also be consumed as a vegetable [18].

In recent years, there has been a significant and steady increase in demand for *E. senticosus* in both domestic and international markets [19]. However, the current production of *E. senticosus* cannot meet the market demand due to its limited cultivation area and low rates of quality seed and yield [20]. The market supply of certain *E. senticosus* products heavily relies on natural forest resources, and the related industries are still in a state of wild or semi-wild utilization [21]. The excessive harvesting of wild resources of *E. senticosus* has led to the depletion of its natural population, and the enormous demand and consumption have made the supply–demand contradiction increasingly prominent. Therefore, the targeted protection and rational utilization of *E. senticosus* germplasm resources, as well as the development, selection, and promotion of superior breeding materials, are of great significance and need to be carried out. In production, *E. senticosus* is mainly propagated by seeds. Because of its slow growth in the seedling stage and long reproductive cycle, it usually takes more than three years to flower and bear fruit or to be harvested directly for medicinal use. Thus, the evaluation and selection of its superior germplasm resources becomes particularly important. However, to date, there has been relatively little research on the breeding program of *E. senticosus*.

The sustainable utilization of germplasm resources relies on an adequate knowledge of the magnitude and pattern of provenance variations. Generally, populations within the same regions of provenance are expected to derive from the same base population or random mating [22,23]. Thus, provenance testing can be employed to identify the genetic variations among populations from different provenances and, exploited through the selection of superior populations for reproductive materials. In a species like *E. senticosus* for which there is no a priori genetic information, variability studies are of fundamental importance to screen the available genetic variation for higher productivity and future genetic improvement works. The selection of best provenances of a given species for a given region is crucial for achieving maximum productivity in plantations. Thus, the aims of this study were to (a) quantify variations in growth, leaf, and photosynthetic traits, as well as in medicinal substance content among 13 provenances of 3-year-old *E. senticosus* and (b) to select fast-growing genotypes that have a good content of medicinal substances using a multi-trait comprehensive evaluation method. The study will provide an empirical basis for the selection and evaluation of superior *E. senticosus* provenance, as well as a reference for future breeding work of *E. senticosus*.

## 2. Materials and Methods

### 2.1. Study Area

Germplasm was collected from 13 provenances of *E. senticosus* in its natural distribution area in 2018 to produce seedlings for field gene bank. The experimental forest is located in the Daliangzi River national improved varieties base for *E. senticosus* and *Aralia elata* of Tangyuan County, Jiamusi City, Heilongjiang Province (46°54′23.6″ N, 129°46′55.3″ E) at an average elevation of 450 m. The plantation was established in a randomized complete block design with double-row arrangement, where each provenance had 50 seedlings and a plant spacing of 1.0 m × 1.5 m. The region experiences an average annual temperature ranging from −2 °C to 10 °C, with a frost-free period of 136.6 days and an average annual effective accumulated temperature of 2559.5 °C. The annual average precipitation is 536.8 mm, with two-thirds of the rainfall concentrated in the summer and autumn seasons. The soil type is dark brown soil. The geographical and climatic characteristics of 13 *E. senticosus* provenances are listed in Table 1.

**Table 1.** Geo-climatic variables of different *Eleutherococcus senticosus* provenances considered in this study.

| No. | Provenance | Latitude (N) | Longitude (E) | Elevation/m | Annual Mean Temperature/$^\circ$C | Annual Precipitation/mm |
|-----|-----------|-------------|--------------|-------------|------------------------------------|-------------------------|
| 1 | Cuiluan (CL) | 47°59′10.00″ | 128°12′30.31″ | 372 | 0.4 | 600.0 |
| 2 | Huinan (HN) | 42°42′36.74″ | 126°01′23.16″ | 400 | 5.0 | 737.4 |
| 3 | Hongxing (HX) | 49°13′44.01″ | 129°37′03.21″ | 415 | −4.0 | 550.0 |
| 4 | Jian (JA) | 41°16′47.85″ | 126°08′10.54″ | 610 | 6.5 | 900.0 |
| 5 | Linjiang (LJ) | 41°50′41.93″ | 126°58′01.66″ | 406 | 3.0 | 875.0 |
| 6 | Langxiang (LX) | 46°57′43.15″ | 128°54′41.41″ | 362 | 0.5 | 600.0 |
| 7 | Nancha (NC) | 47°06′19.76″ | 129°23′24.65″ | 270 | 1.3 | 625.0 |
| 8 | Qitaihe (QTH) | 45°52′24.13″ | 131°03′12.64″ | 240 | 3.0 | 464.9 |
| 9 | Qingyuan (QY) | 42°09′54.15″ | 124°58′38.18″ | 500 | 5.3 | 806.5 |
| 10 | Shuangyashan (SYS) | 46°42′04.25″ | 131°11′11.35″ | 406 | 4.8 | 540.5 |
| 11 | Tonghua (TH) | 41°42′13.18″ | 125°58′23.00″ | 949 | 5.5 | 870.0 |
| 12 | Tangyuan (TY) | 46°54′23.60″ | 129°46′55.30″ | 450 | 2.0 | 536.8 |
| 13 | Xinqing (XQ) | 48°18′33.72″ | 128°35′55.13″ | 475 | 0.6 | 653.7 |

*2.2. Measurement of Growth and Photosynthetic Traits*

In September 2021, 30 healthy individual trees from each provenance with consistent growth and free from pests and diseases were selected for measurement of tree height (TH), basal diameter (BD), leaf traits, and photosynthetic characteristics. The TH was measured using a measuring tape, while BD was measured by caliper. At the same time, three healthy, mature leaves were harvested from the canopy of the plant and placed in a sealed plastic bag. The bag was then placed in a cooler with ice and transported to the laboratory. Due to the fact that the *E. senticosus* has palmately compound leaves, we measured the overall characteristics of the entire large leaf directly. The fresh weight of the leaves (LFW, g) was measured using a 1/10,000 electronic balance and the total leaf area (LA, cm$^2$) was measured based on black–white scanning image [24]. Thereafter, the leaves were placed in an electric drying oven and dried until a constant weight was achieved, and then the leaf dry weight (LDW, g) and leaf water content (LWC) were determined. The photosynthetic characteristics were measured by the lico-6400 photosynthesis measurement system, including single-leaf photosynthetic rate (Pn, $\mu$mol·m$^{-2}$·s$^{-1}$), transpiration rate (Tr, mol·m$^{-2}$·s$^{-1}$), intercellular CO$_2$ concentration (Ci, $\mu$mol·mol$^{-1}$), water use efficiency (WUE, g/kg), and stomatal conductance (Gs, mol·m$^{-2}$·s$^{-1}$) of plants.

*2.3. Determination of Medicinal Substance Contents*

We randomly selected 15 fruit-bearing plants from each provenance to determine the contents of medicinal substances because the measurement of medicinal substances was time-consuming and costly. Subsequently, a mixed sample of every 5 trees was selected to serve as a single replicate, giving three replicates in total for determining syringin content and total flavonoid content. The content of syringin in *E. senticosus* fruits was measured by ultra performance liquid chromatography [25]. The specific steps were preparation of reference solution, preparation of test sample solution, chromatographic analysis, and assessment of linear relationship. For preparation of reference solution, 1.5 mg standard compound of syringin was weighed and placed in a 10 mL volumetric flask, followed by the addition of methanol to prepare a reference solution. For preparation of test sample solution, the fresh fruits were freeze-dried and then crushed, and the powder passed through a 40 mm mesh sieve. Then, 2.0 g fine powder of *E. senticosus* fruit was weighed precisely and placed in a 50 mL erlenmeyer flask, to which 25 mL of methanol was added and left to soak for 8 h. Then, the sample was extracted using ultrasonic extraction (500 W, 40 kHz) 3 times, each time for 30 min. The extracts were then filtered and concentrated under reduced pressure. Finally, methanol was added to make up to 5 mL in a volumetric flask, shook well, and filtered through 0.22 $\mu$m microporous membrane before analysis. The sample solutions were analyzed by ultra performance liquid chromatography (UPLC).

ACQUITY UPLC BEH C$_{18}$ chromatographic column (2.1 × 50 mm, 1.7 μm) was used. The mobile phase consisted of acetonitrile–water solution (gradient elution: 0 min to 3 min, 5% acetonitrile; 3 min to 5 min, 5% to 15% acetonitrile; 5 min to 6 min, 15% to 75% acetonitrile; 6 min to 8 min, 75% to 95% acetonitrile; 8 min to 9 min, 95% acetonitrile; 9 min to 11 min, 95% to 5% acetonitrile; 11 min to 12 min, 5% acetonitrile), with a flow rate of 0.5 mL/min, column temperature of 35 °C, detection wavelength of 270 nm, and injection volume of 2 μL. Different volumes of reference solution of syringin (0.1, 0.3, 0.6, 1.0, 1.5, and 2 μL) were injected into the chromatographic column to measure the chromatogram. We then established a linear relationship between the injection volume as the explanatory variable and the peak area of chromatogram as the response variable. Using the regression equation developed (Y = 2 × 10$^6$X + 22,304, r = 0.9996), the content of syringin of the samples was determined.

The total flavonoid content in the fruits of *E. senticosus* was determined following the spectrophotometric method described by the authors of reference [26]. To extract total flavonoid, fresh dried *E. senticosus* fruits were crushed into fine powder after sieving through a 40-mesh sieve. Thereafter, we took 1.0 g of the fruit powder and transferred it to 10 mL stoppered test tube, to which was added 10 mL of 90% methanol, and let it soak overnight before subjecting it to ultrasonic extraction (500 W, 40 kHz) three times for 30 min each time. After merging the extracts, the mixture was filtered, and finally concentrated under reduced pressure. We then added 90% methanol to the concentrated solution to obtain 25 mL volumetric flask and measured the absorbance at 420 nm. For determining the total flavonoid content in the samples, we developed standard curve (A = 0.0046C + 0.0039, r = 0.9987) using rutin as reference standard.

### 2.4. Statistical Analyses

Variations in growth, leaf, photosynthetics, and contents of medicinal substances among provenances were analyzed using the following linear model, and the significance of the analysis of variance (ANOVA) was estimated using an *F*-test [27]:

$$X_{ij} = \mu + P_i + \varepsilon_{ij} \tag{1}$$

where the $y_{ij}$ was the performance of an individual tree (*j*) in *i*th provenance, $\mu$ was the overall mean value, $P_i$ was the random effect of *i*th provenance, and $\varepsilon_{ij}$ referred to random errors.

The phenotypic coefficient of variation (PCV) was calculated using the following formula [28]:

$$PCV = \frac{\sqrt{\sigma_P^2}}{X} \times 100\% \tag{2}$$

where $\sigma_P^2$ represented the phenotype variance component for the trait and $X$ represented the phenotypic mean value for the trait.

The narrow-sense heritability ($h^2$) was estimated by the following formula [29]:

$$h^2 = \frac{\sigma_A}{\sigma_P} \tag{3}$$

where $\sigma_A$ represented additive variance and $\sigma_P$ represented phenotypic variance.

Phenotypic correlation analysis was performed using the following formula [30]:

$$r_A(xy) = \frac{Cov_{P(xy)}}{\sigma_P(x)\sigma_P(y)} \tag{4}$$

where $Cov_{P(xy)}$ represented the phenotypic covariance between trait $x$ and $y$, $\sigma_P(x)$ represented the phenotypic variance for trait $x$, and $\sigma_P(y)$ represented the phenotypic variance for trait $y$.

The multiple-traits comprehensive evaluation was analyzed from the following formula [31]:

$$a_i = x_{ij}/x_{j\max} \tag{5}$$

$$Qi = \sqrt{\sum_{i=1}^{n} a_i} \tag{6}$$

where $x_{ij}$ represented the average value of the *i*-th provenance for the *j*-th trait, $x_{j\max}$ represented the optimal value of the mean for that trait, $Qi$ represented the comprehensive evaluation value, and *n* represented the number of evaluated traits.

Genetic gain was estimated from the formula [32]:

$$\Delta G = h^2 S / X \tag{7}$$

where $h^2$, *S*, and *X* represented narrow-sense heritability, selection differential, and *X* mean value of the given trait, respectively. All statistical analyses were conducted by using SPSS 27.0 software (IBM SPSS Statistics) and Microsoft Office Excel 2021.

## 3. Results

### 3.1. Variations in Growth and Leaf Traits

Significant differences in total height and all leaf traits were detected among provenances; however, basal diameter did not show significant differences among provenances (Table 2). The mean values of growth and leaf traits of all *E. senticosus* provenances are shown in Table 3. The average TH for all provenances was 111.58 cm, ranging from 99.00 cm (QTH) to 145.67 cm (JA). The average BD was 11.64 mm, ranging from 9.78 mm (QTH) to 13.25 mm (XQ). For leaf traits, the average LA was 143.60 cm$^2$, ranging from 121.10 cm$^2$ (TY) to 173.24 cm$^2$ (SYS). The average LFW was 3.80 g, ranging from 3.13 g (QTH) to 4.57 g (TH); the average LDW was 1.36 g, ranging from 1.10 g (QTH) to 1.66 g (TH); and the average LWC was 64.4%, ranging from 61.1% (XQ) to 66.9% (NC). The phenotypic coefficient of variation was larger than 20.0% for all growth and leaf traits, and, except for TH and LA, all other growth and leaf traits reached moderate levels of narrow-sense heritability. The variance components of provenance were between 66.0 and 99.6%, among which TH, BD, LA, LFW, LDW, and LWC were all above 50% (Table 3).

**Table 2.** Variance analysis of different characteristics among provenances of *E. senticosus*.

| Trait | df | MS | *F* | Significance |
|-------|-----|---------|---------|--------------|
| TH | 12 | 1747.373 | 3.391 | 0.000 |
| BD | 12 | 8.420 | 1.695 | 0.078 |
| LA | 12 | 2768.410 | 3.320 | 0.000 |
| LFW | 12 | 1.715 | 1.945 | 0.037 |
| LDW | 12 | 0.278 | 2.096 | 0.023 |
| LWC | 12 | 0.002 | 2.220 | 0.016 |
| Ci | 12 | 2981.929 | 4.458 | 0.000 |
| Gs | 12 | 8495.118 | 3.142 | 0.001 |
| Pn | 12 | 13.680 | 4.939 | 0.000 |
| Tr | 12 | 2.772 | 4.617 | 0.000 |
| WUE | 12 | 1.169 | 7.094 | 0.000 |
| TFC | 12 | 6.942 | 85.100 | 0.000 |
| SC | 12 | 0.029 | 225.280 | 0.000 |

Notes: TH, tree height. BD, basal diameter. LA, leaf area. LFW, leaf fresh weight. LDW, leaf dry weight. LWC, leaf water content. Ci, intercellular CO$_2$ concentration. Gs, stomatal conductance. Pn, photosynthetic rate. Tr, transpiration rate. WUE, water use efficiency. TFC, total flavonoid content. SC, syringin content. df, degree of freedom. MS, mean square.

### 3.2. Variations in Photosynthetic Characteristics

Photosynthetic traits varied significantly among provenances (Table 2). Table 3 shows the mean values of all photosynthetic traits. The average Pn was 13.13 μmol·m$^{-2}$·s$^{-1}$, ranging from 10.83 μmol·m$^{-2}$·s$^{-1}$ in JA to 15.00 μmol·m$^{-2}$·s$^{-1}$ in TY. The maximum mean value for photosynthetic rate (Pn) was 1.39 times greater than the minimum mean value. The average Ci was 242.75 μmol/mol, ranging from 209.22 μmol/mol (JA) to 274.78 μmol/mol (HX). The average Gs was 184.48 mol·m$^{-2}$·s$^{-1}$, ranging from 113.00 mol·m$^{-2}$·s$^{-1}$ (JA) to

228.78 mol·m$^{-2}$·s$^{-1}$ (QY). The average Tr was 4.44 mol·m$^{-2}$·s$^{-1}$, ranging from 3.29 mol·m$^{-2}$·s$^{-1}$ (HX) to 5.27 mol·m$^{-2}$·s$^{-1}$ (QTH). The average WUE was 3.03 g/kg, ranging from 2.57 g/kg (NC) to 3.86 g/kg (HX). Stomatal conductance (Gs) showed the largest phenotypic variation, while intercellular $CO_2$ concentration (Ci) showed the lowest phenotypic variability, with narrow-sense heritability greater than 20% for all photosynthetic traits (Table 3).

**Table 3.** Mean values of different traits for each provenance together with overall means, standard deviation (SD) phenotypic coefficient of variation (PCV), and narrow-sense heritability.

| Provenance | Growth and Leaf Traits | | | | | | Photosynthetic Traits | | | | | Medicinal Substance | |
|---|---|---|---|---|---|---|---|---|---|---|---|---|---|
| | TH (cm) | BD (mm) | LA (cm$^2$) | LFW (g) | LDW (g) | LWC (%) | Ci (μmol/mol) | Gs (mol·m$^{-2}$·s$^{-1}$) | Pn (μmol·m$^{-2}$·s$^{-1}$) | Tr (mol·m$^{-2}$·s$^{-1}$) | WUE (g/kg) | TFC (mg/g) | SC (mg/g) |
| CL | 126.78 | 11.90 | 131.09 | 3.45 | 1.20 | 65.2 | 265.33 | 204.00 | 13.34 | 4.85 | 2.77 | 4.441 | 0.332 |
| HN | 115.44 | 10.98 | 126.43 | 3.91 | 1.38 | 65.0 | 242.00 | 189.56 | 13.86 | 4.82 | 2.92 | 3.131 | 0.33 |
| HX | 107.11 | 12.66 | 166.32 | 3.92 | 1.38 | 65.0 | 274.78 | 182.56 | 11.96 | 3.29 | 3.86 | 0.781 | 0.184 |
| JA | 145.67 | 12.24 | 154.26 | 4.11 | 1.51 | 63.8 | 209.22 | 113.00 | 10.83 | 3.57 | 3.02 | 3.644 | 0.235 |
| LJ | 104.44 | 11.39 | 129.15 | 3.39 | 1.20 | 64.7 | 242.56 | 199.78 | 14.52 | 4.47 | 3.30 | 0.809 | 0.197 |
| LX | 111.00 | 11.20 | 150.15 | 4.18 | 1.42 | 65.9 | 230.67 | 175.00 | 12.90 | 4.54 | 2.88 | 0.734 | 0.215 |
| NC | 125.67 | 11.66 | 137.10 | 3.45 | 1.15 | 66.9 | 235.78 | 192.11 | 11.90 | 4.78 | 2.57 | 0.213 | 0.109 |
| QTH | 99.00 | 9.78 | 121.53 | 3.13 | 1.10 | 64.8 | 245.56 | 193.33 | 13.41 | 5.27 | 2.57 | 0.586 | 0.153 |
| QY | 113.33 | 11.94 | 141.79 | 3.81 | 1.41 | 62.8 | 239.78 | 228.78 | 14.73 | 4.24 | 3.54 | 3.116 | 0.366 |
| SYS | 104.89 | 12.80 | 173.24 | 4.26 | 1.50 | 64.8 | 226.33 | 150.11 | 12.33 | 4.11 | 3.01 | 0.68 | 0.154 |
| TH | 123.33 | 10.63 | 163.04 | 4.57 | 1.66 | 63.8 | 225.56 | 166.67 | 13.53 | 4.59 | 2.95 | 4.08 | 0.255 |
| TY | 126.33 | 10.89 | 121.1 | 3.23 | 1.18 | 63.3 | 260.44 | 227.11 | 15.00 | 4.97 | 3.02 | 2.921 | 0.264 |
| XQ | 138.56 | 13.25 | 151.57 | 4.01 | 1.56 | 61.1 | 257.78 | 176.22 | 12.38 | 4.22 | 2.97 | 2.368 | 0.458 |
| Mean | 118.58 | 11.64 | 143.60 | 3.80 | 1.36 | 64.4 | 242.75 | 184.48 | 13.13 | 4.44 | 3.03 | 2.116 | 0.250 |
| SD | 25.35 | 2.31 | 32.16 | 0.98 | 0.38 | 0.03 | 30.14 | 57.47 | 1.97 | 0.91 | 0.52 | 1.500 | 0.100 |
| PCV | 21.4 | 19.8 | 22.4 | 25.9 | 28.3 | 4.8 | 12.4 | 31.2 | 15.0 | 20.5 | 17.1 | 70.9 | 38.7 |
| Heritability | 0.28 ± 0.20 | 0.39 ± 0.14 | 0.28 ± 0.19 | 0.37 ± 0.15 | 0.36 ± 0.15 | 0.32 ± 0.15 | 0.25 ± 0.22 | 0.29 ± 0.0.19 | 0.23 ± 0.22 | 0.24 ± 0.22 | 0.20 ± 0.23 | 0.31 ± 0.03 | 0.31 ± 0.01 |
| Variance component | 77.2 | 62.9 | 76.9 | 66.0 | 67.6 | 66.7 | 81.7 | 75.9 | 83.2 | 82.2 | 87.6 | 98.8 | 99.6 |

Notes: TH, tree height. BD, basal diameter. LA, leaf area. LFW, leaf fresh weight. LDW, leaf dry weight. LWC, leaf water content. Ci, intercellular $CO_2$ concentration. Gs, stomatal conductance. Pn, photosynthetic rate. Tr, transpiration rate. WUE, water use efficiency. SC, syringin content. TFC, total flavonoid content. SD, standard deviation. PCV, phenotypic coefficient of variation. Heritability, narrow-sense heritability. Variance component (%), the percentage of provenance variance in the total variance.

### 3.3. Variations in Contents of Medicinal Substances

Highly significant variations were observed in the contents of medicinal substances among provenances (Table 2). The average syringin content in fresh fruits of *E. senticosus* was 0.25 mg/g, ranging from 0.109 mg/g to 0.458 mg/g (Table 3). The syringin content of XQ provenance with the highest concentration was 4.2 times higher than that of NC provenance with the lowest concentration. The average total flavonoid content was 2.116 mg/g, ranging from 0.213 mg/g to 4.441 mg/g (Table 3). The provenance with the largest total flavonoid content (CL) had 20.9 times more total flavonoid than the provenance with the lowest total flavonoid content (NC). The total flavonoid content exhibited larger phenotypic variability than syringin content, but the narrow-sense heritability of both traits had more than 0.3. Meanwhile, the variance component of TFC and SC reached 98.8% and 99.6%, respectively (Table 3).

### 3.4. Intercharacter Correlations

Significant correlations were found between growth and leaf traits, photosynthetic characters, and contents of medicinal substances (Table 4). To investigate the relationship between various growth traits and the content of medicinal substances in different provenances of *E. senticosus*, we conducted a correlation analysis of 13 traits (Table 4). Relatively weak correlations were observed between TH and BD ($r = 0.353$, $p < 0.05$), and BD and LA ($r = 0.381$, $p < 0.05$), while relatively strong correlations were observed between LA and LFW ($r = 0.764$, $p < 0.01$), LA and LDW ($r = 0.709$, $p < 0.01$), and LFW and LDW ($r = 0.938$, $p < 0.01$). There were no significant correlations between photosynthetic characters and growth and leaf traits. However, photosynthetic traits showed correlations among themselves, such as between Ci, Gs, Tr, and Pn ($r = 0.362 \sim 0.775$, $p < 0.05$), Gs, Pn, and Tr ($r = 0.694 \sim 0.746$, $p < 0.01$), and Tr and WUE ($r = -0.716$, $p < 0.01$). While total flavonoid content showed a positive correlation with total height and a negative correlation with

leaf water content, the syringin content showed a significant positive correlation with total height only. A significant positive correlation was also observed between the contents of medicinal substances.

**Table 4.** Correlation analysis of various traits of *E. senticosus* provenances.

| Trait | TH | BD | LA | LFW | LDW | LWC | Ci | Gs | Pn | Tr | WUE | TFC |
|---|---|---|---|---|---|---|---|---|---|---|---|---|
| BD | 0.353 * | | | | | | | | | | | |
| LA | 0.198 | 0.381 * | | | | | | | | | | |
| LFW | 0.127 | 0.315 | 0.764 ** | | | | | | | | | |
| LDW | 0.150 | 0.263 | 0.709 ** | 0.938 ** | | | | | | | | |
| LWC | −0.093 | 0.006 | −0.091 | −0.157 | −0.485 ** | | | | | | | |
| Ci | 0.082 | 0.042 | −0.051 | −0.257 | −0.264 | 0.076 | | | | | | |
| Gs | 0.030 | −0.145 | −0.095 | −0.218 | −0.255 | 0.135 | 0.775 ** | | | | | |
| Pn | −0.083 | −0.175 | −0.066 | −0.025 | −0.051 | 0.022 | 0.349 * | 0.746 ** | | | | |
| Tr | 0.028 | −0.312 | −0.218 | −0.216 | −0.290 | 0.245 | 0.362 * | 0.694 ** | 0.671 ** | | | |
| WUE | −0.146 | 0.286 | 0.231 | 0.195 | 0.245 | −0.218 | −0.018 | −0.167 | −0.002 | −0.716 ** | | |
| TFC | 0.333 * | 0.161 | −0.074 | 0.103 | 0.249 | −0.445 ** | 0.227 | 0.174 | 0.216 | 0.004 | 0.116 | |
| SC | 0.423 ** | 0.033 | −0.036 | 0.191 | 0.271 | −0.262 | −0.022 | 0.006 | 0.175 | 0.043 | −0.005 | 0.649 ** |

Notes: Correlation significance: * $p < 0.05$ (1-tailed), significant; ** $p < 0.01$ (2-tailed), very significant. TH, tree height. BD, basal diameter. LA, leaf area. LFW, leaf fresh weight. LDW, leaf dry weight. LWC, leaf water content. Ci, intercellular $CO_2$ concentration. Gs, stomatal conductance. Pn, photosynthetic rate. Tr, transpiration rate. WUE, water use efficiency. SC, syringin content. TFC, total flavonoid content.

### 3.5. Selection of Superior Provenance of E. senticosus

Principal component analysis of 13 traits was performed to summarize the variation in few latent variables, which will allow the selection of superior provenances based on the comprehensive evaluation method. A total of five principal components were extracted, with the cumulative contribution rate reaching 83.4% (Table 5). The first principal component had an eigenvalue of 3.781 and a contribution rate of 29.1%, of which the eigenvalue of LA (0.637), LFW (0.744), and LDW (0.807) were relatively high, indicating that principal component I mainly represented leaf traits. The second principal component had an eigenvalue of 2.669 and a contribution rate of 20.5%, with relatively high absolute eigenvalues for TFC (0.704) and SC (0.61), indicating that principal component II mainly represented medicinal substances traits. The fifth principal component had a relatively high absolute eigenvalue for TH (0.645) and BD (0.636), indicating that it represented growth traits. Therefore, these three principal components could reflect most of the information on the traits of the 13 provenances of *E. senticosus*, and can provide a reference for the subsequent selection of the superior provenance.

Based on the results of correlation analysis and principal component analysis, we evaluated and calculated the Qi values (Table 6) for 13 provenances of *E. senticosus* using growth traits (TH and BD), leaf traits (LA, LFW and LDW), photosynthetic traits (Pn, Ci, Gs and Tr), and medicinal substances traits (SC and TFC) as selection indexes. Consequently, using growth traits with a selection rate of 10%, it was possible to screen and identify two superior provenances, JA and XQ. The average TH of the selected superior provenances were 27.09 cm (JA) and 19.98 cm (XQ) higher than the overall mean. The genetic gain was 5.6%. A selection according to the medicinal traits identified CL and XQ as superior provenances, with the genetic gains for TFC and SC being 18.9% and 18.0%, respectively. Combining growth and medicinal traits, XQ and QY provenances were selected as superior provenances, and the genetic gains for TH, TFC, and SC were 1.7%, 9.2%, and 20.1%, respectively.

**Table 5.** Results of principal component analysis of various traits of *E. senticosus*.

| Principal Component Analysis | Principal Component I | Principal Component II | Principal Component III | Principal Component IV | Principal Component V |
|---|---|---|---|---|---|
| Eigenvalue | 3.781 | 2.669 | 1.699 | 1.393 | 1.305 |
| Contribution rate | 29.1 | 20.5 | 13.1 | 10.7 | 10.0 |
| Cumulative contribution rate | 29.1 | 49.6 | 62.7 | 73.4 | 83.4 |
| TH | 0.202 | 0.453 | −0.240 | −0.275 | 0.645 |
| BD | 0.473 | 0.168 | 0.119 | 0.268 | 0.636 |
| LA | 0.637 | 0.252 | 0.614 | −0.029 | 0.100 |
| LFW | 0.744 | 0.329 | 0.458 | −0.216 | −0.131 |
| LDW | 0.807 | 0.395 | 0.251 | −0.165 | −0.267 |
| LWC | −0.403 | −0.325 | 0.403 | −0.108 | 0.437 |
| Ci | −0.488 | 0.475 | 0.147 | 0.477 | 0.248 |
| Gs | −0.664 | 0.597 | 0.295 | 0.248 | 0.002 |
| Pn | −0.471 | 0.617 | 0.259 | 0.123 | −0.331 |
| Tr | −0.743 | 0.415 | 0.262 | −0.408 | −0.053 |
| WUE | 0.486 | −0.034 | −0.038 | 0.783 | −0.147 |
| TFC | 0.144 | 0.704 | −0.542 | 0.118 | −0.029 |
| SC | 0.190 | 0.610 | −0.524 | −0.246 | −0.018 |

**Table 6.** Qi values of different provenances of *E. senticosus*.

| Growth Trait | | Medicinal Trait | | Growth and Medicinal Trait | |
|---|---|---|---|---|---|
| Provenance | Qi | Provenance | Qi | Provenance | Qi |
| JA | 1.927 | CL | 1.313 | XQ | 3.207 |
| XQ | 1.925 | XQ | 1.238 | QY | 3.203 |
| SYS | 1.903 | QY | 1.225 | TH | 3.187 |
| TH | 1.895 | TH | 1.215 | CL | 3.180 |
| HX | 1.874 | HN | 1.194 | HN | 3.125 |
| LX | 1.841 | JA | 1.154 | TY | 3.116 |
| QY | 1.825 | TY | 1.110 | JA | 3.065 |
| NC | 1.814 | LX | 0.797 | HX | 3.023 |
| CL | 1.812 | LJ | 0.783 | LX | 2.992 |
| HN | 1.791 | HX | 0.760 | SYS | 2.979 |
| TY | 1.759 | SYS | 0.699 | LJ | 2.971 |
| LJ | 1.751 | QTH | 0.682 | NC | 2.886 |
| QTH | 1.675 | NC | 0.534 | QTH | 2.870 |

## 4. Discussion

The ample genetic diversity present in forest germplasm resources serves as the fundamental basis for genetic breeding [33]. These resources have been shaped over an extended period through both natural and artificial selection processes, and harbor a vast diversity of genes [34]. Consequently, they are an essential source material necessary for variety selection and breeding research. Only by utilizing ideal germplasm resources can novel, superior cultivars be developed through the application of cutting-edge technologies [35,36]. Currently, there has been a lack of systematic research conducted for the collection, evaluation, and selection breeding of *E. senticosus* germplasm resources. In our study, we extensively gathered 13 natural germplasm resources of *E. senticosus* and examined the variation in important traits as the first step for genetic improvement in the species to screen out superior breeding materials. An analysis of variance revealed significant variations in all studied traits except basal diameter. These results indicated that there exist rich variations among the different sources of *E. senticosus*. Therefore, there is a great potential for improving this species, and evaluating and selecting superior provenances is meaningful. The existence of large variations among natural populations and provenances has been observed for many

other tree species, such as *Pinus halepensis* Mill. [37], *Cunninghamia lanceolata* (Lamb.) [38], *Khaya senegalensis* A. Juss [39], and *Cordia africana* Lam [40].

The phenotypic coefficient of variation is an important indicator in the evaluation and selection of high-quality materials for experimentation [41]. Typically, materials with a larger degree of variability are selected for such purposes (less than 10% is low PCV, 10–20% is moderate, and more than 20% is high). In this study, the range of phenotypic coefficient of variation was found to be between 4.8% and 70.9%. Notably, the highest coefficients of variation were observed for the medicinal traits (TFC and SC), with values reaching as high as 70.9% and 38.7%, respectively. Thus, using medicinal properties as an evaluation criterion holds greater potential for the selection of superior provenances.

The traits of plants are determined by the combined effects of genetic and environmental factors. Heritability represents the proportion of a plant's traits that are controlled by genetics. A low heritability ranges from 0 to 0.3, a moderate heritability ranges from 0.31 to 0.6, and a high heritability ranges between 0.61 and 1 [42]. A higher heritability signifies that a particular trait is less affected by the environment and can be more stably passed down to progeny [43]. In a study on the analysis of variation in 53 half-sibling families of *Pinus koraiensis* [44], it was found that some traits (such as ground diameter, breast diameter, and wood volume) had narrow-sense heritability greater than 0.7, indicating high heritability. This suggests that the selected material's traits are relatively stable in inheritance, which is beneficial for evaluating and selecting superior family members. In our study, the narrow-sense heritability of 13 different *E. senticosus* provenances were all above 0.2, with DB, LFW, LDW, LWC, TFC, and SC having a heritability exceeding 0.3, placing them at a moderate level. The aforementioned results indicate that genetic factors play a significant role in the inheritance of the traits observed in *E. senticosus*. These findings provide a foundation for the further selection of superior germplasms characterized by stable phenotypes.

Correlation analysis can aid in understanding the degree of relationship between various traits [45]. In this study, we found positive correlations between total height and contents of medicinal substances, suggesting that the selection of one character simultaneously improves the other traits. Han et al. [12] found that the content of medicinal substances in *E. senticosus* increased with the length of planting years, exhibiting a growth trend until approximately 5 years later when secondary metabolite production no longer showed significant increases. These findings highlight the influence of the duration of cultivation on the secondary metabolite accumulation in *E. senticosus* and suggest a potential optimal period for its cultivation in order to increase its medicinal properties. Therefore, it can be demonstrated that faster growth rate and higher content of medicinal compounds can serve as exemplary criteria for selecting superior provenance in the breeding research of *E. senticosus*. Similarly, in the selection of superior germplasm in other tree species, a comprehensive evaluation of multiple traits has also been employed. For example, Xu et al. [46] used a combination of growth traits and trunk traits to breed superior *Populus* clones, while Liu et al. [47] used fruit traits and oil content to select high-yielding *Idesia polycarpa* with high oil content. These studies have provided excellent material foundations for the genetic improvement of relevant tree species.

Since the whole plant of *E. senticosus* can be used for medicinal purposes, the selection of germplasm resources with high growth rate and high medicinal substance content is highly desirable. In this regard, we integrated growth traits and medicinal properties as selection criteria and preliminarily selected two superior provenances, XQ and QY. Strikingly, both provenances displayed remarkable genetic gains in TH, TFC, and SC, with gains of 1.7%, 9.2%, and 20.1%, respectively. It is worth noting that the selected XQ provenances exceeded the overall average by 16.9%, 11.9%, and 83.2%, respectively, in TH, TFC, and SC traits. These selected superior provenances exhibited advantages in both growth and medicinal substance content, and could be employed for afforestation to enhance economic benefits. In addition, these provenances can be used as base populations for future im-

provement research to increase the quality and quantity of medicinal substances extracted from the species.

## 5. Conclusions

In our results, significant variations were observed among 13 different provenances of *E. senticosus*, with rich genetic variation among the various provenances. And the narrow-sense heritability of different traits in different provenances were all above 0.2. Furthermore, we performed a combined evaluation of different *E. senticosus* provenances based on correlation analysis, principal component analysis, and Qi value results, and comprehensively selected two superior germplasms (XQ and QY provenances) with higher productivity and better stability. Our study provides sufficient empirical basis for the selection and evaluation of superior germplasms in *E. senticosus*, and also provides the basic materials for future genetic improvement work in *E. senticosus*.

**Author Contributions:** S.Z., X.Y. and T.S. planned and designed the research. X.Y. and T.S. obtained the funding acquisition. S.Z. wrote the first draft. S.Z. and H.Z. analyzed data. S.Z., H.Z., L.D., Y.X., W.D. and X.H. performed the experiments and fieldwork. X.Y. and T.S. reviewed and edited the manuscript. All authors have read and agreed to the published version of the manuscript.

**Funding:** This research was funded by "Research Funds Project of Provincial Research Institutes of Heilongjiang Province, project number CZKYF2023-1-A007" and "Dean Fund Project of Heilongjiang Academy of Sciences, project number YZ2023YZNY01". And the APC was funded by the "Research Funds Project of Provincial Research Institutes of Heilongjiang Province, project number CZKYF2023-1-A007".

**Data Availability Statement:** The processed data required to reproduce these findings cannot be shared at this time as the data also forms part of an ongoing study.

**Acknowledgments:** We would like to express our deepest gratitude to Mulualem Tigabu from the Swedish University of Agricultural Sciences for editing the language of our manuscript and providing invaluable revision suggestions. We would also like to extend heartfelt thanks to Guanzheng Qu from Northeast Forestry University, and Xiyang Zhao from Jilin Agricultural University, for their assistance in collecting experimental materials and reviewing our manuscript.

**Conflicts of Interest:** The authors declare no conflict of interest.

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
