# Peer review of "Evaluation and Selection of Excellent Provenances of Eleutherococcus senticosus"

_forests, doi:10.3390/f14071359_

Round 1

Reviewer 1 Report

Dear Authors.

Authors commend that provenance test of Eleutherococcus senticosus show significant differences in growth and medicinal properties among provenances, and that selection of good provenances for both traits could contribute in operational cultivation for E. senticosus industry. Reviewer regards this article as a useful one for E. senticousus research and highly evaluate the paper as a very significant result and believe it deserves to be included in Forests. however, as for the contents of this paper, there are no major problems, there are several minor problems such as discrepancies between figures in text and tables and the lack of adequate explanation in Material and Methods. In addition, in spite of using seedling in this provenance test, authors use analysis technique of broad-sense heritability to estimate inheritance and genetic gain.  Reviewer recommends narrow-sense heritability for this study.

Detailed revisions are listed below:

Abstract

P1L30, P1L35,36

Numerical values with % should be written to one decimal place(4.8314.8, 70.86870.9, 4.384.4, 29.2329.2, 65.5465.5).

Materials and Methods

P4L167

YijXij or correct equation (1), XijYij

P4L174

Why is broad-sense heritability used? Authors conducted this test by seedlings, not clones. So, you should adopt a narrow-sense heritability to estimate the degree of inheritance and genetic gain.

P4L182

The explanation of equation (5) is not sufficient, especially there is no explanation of suffix i and j in Xij. Does Xij mean the average value of the j-th provenance for the i-th trait? Please explain in detail.

Results

P5L195,196

Compared to the other traits, the name of provenances is not shown with maximum and minimum values for basal diameter. Authors showed the name of provenances.

P5L198

Average LDW is 1.358 in the sentence but the value in Table 3 is 1.36.  Reviewer recommends the latter value because other values in this paragraph are two decimal place.

P6L210

Average Pn is 13.131 in the sentence but the value in Table 3 is 13.13.  Reviewer recommends the latter value because other values in this paragraph are two decimal place.

P6L213

The value of average Ci is 243.75 in the text but the value in the Table 3 is 242.75.  Which is true?  Authors must correct the number that is wrong.

P6L215

Average Tr is 4.441 in the sentence but the value in Table 3 is 4.44.  Reviewer recommends the latter value because other values in this paragraph are two decimal place.

P7L232

Average total flavonoid content is 2.116 in the sentence but the value in Table 3 is 2.12.  Reviewer recommends the former value because values in the Table are all three decimal place in TFC.

P8L263, 264, 26736

Numerical values should be written to one decimal place(83.44683.4, 29.08829.1, 20.53220.5) because these values exhibit in percentage. Also you should correct these values in Table 5.

P8L269~270

Eigenvalue of TH and BD in the third principal component in the text are different from those of Table 5. Those of the fifth principal component in the table is the same in the text.  Which is correct? 

Discussion

P9L311

Numerical values should be written to one decimal place(4.8314.8, 70.86870.9).

P9L320

The four traits listed here are over 0.7, so if you should express 0.7, not 0.61. Also there are other traits that exceed 0.7 (Ci, Tr and WUE), but why are only these four traits listed here? You should express Ci, Tr, WUE in the sentence that are all over 0.7 in heritability.

Tables

Table1

One of provenance (No.13) is located in Russia according to the latitude and longitude, is this correct? I think that 128 degrees longitude is correct, not 138 degrees if the name of provenance is correct. Please check the exact location.

Table2

Authors can delete variance source from the Table because of only one term (provenance).

Table3

Average of Ci is different from that value of the sentence (P6L213).  Authors must correct the number that is wrong.

Table5

Eigenvalue of TH and BD in the third principal component in the text are different from those of Table 5. Those of the fifth principal component in the table is the same in the text.  Which is correct?

Author Response

List of response

Manuscript ID: forests-2443588

Title: Evaluation and selection of excellent provenances of Eleutherococcus senticosus

Dear Editor-in-chief and Reviewers:

Thanks for your letter and for the reviewers’ comments concerning our manuscript entitled "Evaluation and selection of excellent provenances of Eleutherococcus senticosus". Those comments are all valuable and very helpful for revising and improving the quality our paper. We have studied all comments carefully and have made corrections which we hope meet the standard. For ease of tracking, we uploaded files with track change and clean copy. We hope that the revision is satisfactory and we look forward to hearing from you. If there have any further questions, please feel free to contact us and we will make changes immediately.

The main corrections in the paper and the responses to the reviewers’ comments are given below.

Responds to reviewer’s comments:

Reviewer 1

Detailed revisions are listed below:

Abstract

P1L30, P1L35,36

Numerical values with % should be written to one decimal place(4.831→4.8, 70.868→70.9, 4.38→4.4, 29.23→29.2, 65.54→65.5).

Response: Thank you very much for your detailed comments, we have revised the wrong way of writing the decimal point in the article (4.831%→4.8%, 70.868%→70.9%, 4.38%→4.4%, 29.23%→29.2%, 65.54%→65.5%). See revised version manuscript for details of where to change.

Materials and Methods

P4L167

Yij→Xij or correct equation (1), Xij→Yij

Response: Thanks for the detailed comments, we have revised this mistake. See revised version manuscript for details of where to change.

P4L174

Why is broad-sense heritability used? Authors conducted this test by seedlings, not clones. So, you should adopt a narrow-sense heritability to estimate the degree of inheritance and genetic gain.

Response: Dear reviewer, we appreciate the valuable comment by the reviewer 1. We have given full consideration to the question you raised. We recalculated narrow-sense heritability to estimate the degree of inheritance and genetic gain.

At the same time, we also synthesized the opinions of reviewer 2. He said: “In principle, heritability does not add much to this study only unnecessarily complicating the inference. As an alternative, the percentage of provenance variance in the total variance can be given”.

So, we took both opinions into account and decided to split the difference. We also calculated the variance component and it is listed in Table 3.

P4L182

The explanation of equation (5) is not sufficient, especially there is no explanation of suffix i and j in Xij. Does Xij mean the average value of the j-th provenance for the i-th trait? Please explain in detail.

Response: Thanks for the detailed comments, we have re-added a detailed explanation of this formula.

Results

P5L195,196

Compared to the other traits, the name of provenances is not shown with maximum and minimum values for basal diameter. Authors showed the name of provenances.

Response: Thanks for the detailed comments, we have re-added the provenance name of the maximum and minimum values in basal diameter trait.

P5L198

Average LDW is 1.358 in the sentence but the value in Table 3 is 1.36.  Reviewer recommends the latter value because other values in this paragraph are two decimal place.

Response: Thanks for the detailed comments, we have revised this mistake and all decimal values have been changed to only two decimal places.

P6L210

Average Pn is 13.131 in the sentence but the value in Table 3 is 13.13.  Reviewer recommends the latter value because other values in this paragraph are two decimal place.

Response: Thanks for the detailed comments, we have revised this mistake and all decimal values have been changed to only two decimal places.

P6L213

The value of average Ci is 243.75 in the text but the value in the Table 3 is 242.75.  Which is true?  Authors must correct the number that is wrong.

Response: Thanks for the detailed comments, we have revised this mistake. The average Ci is 242.75.

P6L215

Average Tr is 4.441 in the sentence but the value in Table 3 is 4.44.  Reviewer recommends the latter value because other values in this paragraph are two decimal place.

 Response: Thanks for the detailed comments, we have changed the 4.441 to 4.44.

P7L232

Average total flavonoid content is 2.116 in the sentence but the value in Table 3 is 2.12.  Reviewer recommends the former value because values in the Table are all three decimal place in TFC.

 Response: Thanks for your suggestion, we have corrected the error. (See table 3)

P8L263, 264, 26736

Numerical values should be written to one decimal place(83.446→83.4, 29.088→29.1, 20.532→20.5) because these values exhibit in percentage. Also you should correct these values in Table 5.

 Response: Thank you very much for your detailed comments, we have revised the wrong way of writing the decimal point in the article (83.446%→83.4%, 29.088%→29.1%, 20.532%→20.5%) and also corrected these values in Table 5.

P8L269~270

Eigenvalue of TH and BD in the third principal component in the text are different from those of Table 5. Those of the fifth principal component in the table is the same in the text.  Which is correct?

 Response: Thank you very much for your care in finding this problem for us, and now we have resolved the error. Sincere thanks again!

Discussion

P9L311

Numerical values should be written to one decimal place(4.831→4.8, 70.868→70.9).

 Response: Thank you very much for your detailed comments, we have revised the wrong way of writing the decimal point in the article (4.831%→4.8%, 70.868%→70.9%).

P9L320

The four traits listed here are over 0.7, so if you should express 0.7, not 0.61. Also there are other traits that exceed 0.7 (Ci, Tr and WUE), but why are only these four traits listed here? You should express Ci, Tr, WUE in the sentence that are all over 0.7 in heritability.

 Response: Thanks for your valuable comments, we have re-added a more complete description of the results.

Tables

Table1

One of provenance (No.13) is located in Russia according to the latitude and longitude, is this correct? I think that 128 degrees longitude is correct, not 138 degrees if the name of provenance is correct. Please check the exact location.

 Response: Thanks for your valuable comments, the 128 degrees longitude is correct, and we have corrected it. (See table 1)

Table2

Authors can delete variance source from the Table because of only one term (provenance).

 Response: Thanks for your valuable comments, we have deleted variance source from the Table. (See table 2)

Table3

Average of Ci is different from that value of the sentence (P6L213).  Authors must correct the number that is wrong.

 Response: Thanks for your valuable comments, we have revised it. (See table 2)

Table5

Eigenvalue of TH and BD in the third principal component in the text are different from those of Table 5. Those of the fifth principal component in the table is the same in the text.  Which is correct?

Response: Thanks for your valuable comments, the principal component analysis in Table 5 is correct. We have changed the incorrect description in the article.

Reviewer 2 Report

The work describes the variability in growth traits and chemical components of ginseng. The study provides background information on the provenance variability of this commercially used species.

In the introduction, I lacked a description of the specifics of the cultivation of this species, which is probably obvious to the authors but for a reader unfamiliar with the specifics would provide the necessary background for understanding the study.

I have a few methodological comments:

- what material is tested is not mentioned and does the term the germplasm mean seedlings obtained by generative reproduction?

- why was the block effect ignored in the ANOVA analyses, despite the fact that the experiment was set up in a random block design

- it was not justified why heritability was used in a broad sense. This term should in principle not be calculated for provenance only for families or individuals.

- no error in estimating heritability is given.

In principle, heritability does not add much to this study only unnecessarily complicating the inference. As an alternative, the percentage of provenance variance in the total variance can be given.

The discussion of the results is cursory and very general. I propose to expand the discussion by referring to the results obtained. The results indicate low variation in growth traits while high variation in flavonoid and syringin content.

The summary does not highlight "the theoretical basis for the selection and evaluation of superior E. senticosus provenance, as well as a reference for future breeding work of E. senticosus", which was indicated as one of the aims of the work.

 Apart from the above-mentioned remarks, I find the work interesting and brings new information about the provenience variability of E. senticosus.

The language of the work is understandable. However, I noticed a awkward phrases:

"Heritability is another critical parameter in the selective breeding and genetic gain estimation". Selective breeding is too literal a translation I suggest using  "breeding program". In my opinion there is no need to highlight that selection is man-made. There are a few more similar examples in the text but this does not hinder the understanding of the text except for the methodological part.  

Author Response

Dear reviewer,

Thank you very much for your valuable comments. Your recommendation has greatly improved our article. We have revised the article according to your comments, and the relevant revisions have been marked in the revised version of the manuscript. At the same time, we have also uploaded a clean version for your review. Thanks again!

Responds to reviewer’s comments:

Reviewer 2

Comments and Suggestions for Authors:

The work describes the variability in growth traits and chemical components of ginseng. The study provides background information on the provenance variability of this commercially used species.

In the introduction, I lacked a description of the specifics of the cultivation of this species, which is probably obvious to the authors but for a reader unfamiliar with the specifics would provide the necessary background for understanding the study.

Response: Thanks for your valuable comments. We have also made a few changes based on your comments. Our research mainly focuses on the evaluation and selection of E. senticosus provenances. In the introduction, we have explained the reasons why we are conducting this work. In production, the main propagation method of E. senticosus is seed propagation. However, the materials are all from natural forests, which results in slow growth, low fruit yield, and unstable characteristics. Therefore, our research aims to select optimal provenances of E. senticosus with better traits and promote them for planting, hoping to provide excellent breeding materials for the industry of E. senticosus.

I have a few methodological comments:

1- what material is tested is not mentioned and does the term the germplasm mean seedlings obtained by generative reproduction?

Response: Thanks for your valuable comments. Regarding this question, I would like to provide you with the following explanation. Germplasm resources, also known as genetic resources, refer to the genetic material that is transmitted from the parent to offspring of a living organism. Germplasm is often found in specific varieties, including ancient local varieties, newly developed promotion varieties, important genetic materials, and wild relatives of plants. In this study, we collected the roots of Eleutherococcus senticosus from natural forests, propagated them through buried roots, and established a germplasm resource bank. Finally, genetic determination, evaluation and selection were performed.

2- why was the block effect ignored in the ANOVA analyses, despite the fact that the experiment was set up in a random block design

Response: Thanks for your valuable comments. This experiment was based on the germplasm resource bank which established earlier. However, since we did not set up blocks when establishing the resource bank, the block effect was not reflected in this article.

3- it was not justified why heritability was used in a broad sense. This term should in principle not be calculated for provenance only for families or individuals.

Response: Thanks for your valuable comments. We have given full consideration to the question you raised. We recalculated narrow-sense heritability to estimate the degree of inheritance and genetic gain.

4- no error in estimating heritability is given.

Response: Thanks for your valuable comments. We have reconfigured the error in heritability.

In principle, heritability does not add much to this study only unnecessarily complicating the inference. As an alternative, the percentage of provenance variance in the total variance can be given.

Response: Thanks for your valuable comments. The traits of plants are determined by the combined effects of genetic and environmental factors. The concept of "heritability" represents the proportion of a plant's traits that are controlled by genetics. The higher the heritability, the greater the proportion of genetic factors in determining the traits, and the smaller the proportion of environmental factors. The analysis of the heritability of each trait is helpful for us to select traits with higher heritability for comprehensive evaluation in the follow-up study. Of course, we also recalculated the variance components and included them in the results. I hope my answer is to your satisfaction. Thanks again!

The discussion of the results is cursory and very general. I propose to expand the discussion by referring to the results obtained. The results indicate low variation in growth traits while high variation in flavonoid and syringin content.

Response: Thanks for your valuable comments. Breeding studies for the selection of trees have been extensively researched in many species, and many superior varieties have been identified, bringing considerable benefits to production practices. However, E. senticosus, the species studied in this research, has not yet been evaluated and screened for germplasm resources due to the fact that the related industry was entirely based on the exploitation of natural forests before. With the gradual depletion of wild resources, large-scale artificial cultivation has become unavoidable. We could not find any literature related to the evaluation of seed sources of E. senticosus. As an alternative, we reviewed some similar research results on other species as a reference.

The summary does not highlight "the theoretical basis for the selection and evaluation of superior E. senticosus provenance, as well as a reference for future breeding work of E. senticosus", which was indicated as one of the aims of the work.

Response: Thank you for your valuable comments. We have re-added the main idea of this article in the summary part.

 Apart from the above-mentioned remarks, I find the work interesting and brings new information about the provenience variability of E. senticosus.

Response: Thank you again for your valuable comments!

Comments on the Quality of English Language

The language of the work is understandable. However, I noticed a awkward phrases:

"Heritability is another critical parameter in the selective breeding and genetic gain estimation". Selective breeding is too literal a translation I suggest using  "breeding program". In my opinion there is no need to highlight that selection is man-made. There are a few more similar examples in the text but this does not hinder the understanding of the text except for the methodological part.

 Response: Thanks a lot. We have made the necessary modifications to similar sentence in this article.
